# REVersal of nEuromusculAr bLocking Agents in Patients Undergoing General Anaesthesia (REVEAL Study)

**DOI:** 10.3390/jcm12020563

**Published:** 2023-01-10

**Authors:** Massimiliano Greco, Pier Francesco Caruso, Giovanni Angelotti, Romina Aceto, Giacomo Coppalini, Nicolò Martinetti, Marco Albini, Lori D. Bash, Michele Carvello, Federico Piccioni, Roberta Monzani, Marco Montorsi, Maurizio Cecconi

**Affiliations:** 1Department of Biomedical Sciences, Humanitas University, Pieve Emanuele, 20072 Milan, Italy; 2Department of Anaesthesiology and Intensive Care, IRCCS Humanitas Research Hospital, 20089 Milan, Italy; 3IRCCS Humanitas Research Hospital, 20089 Milan, Italy; 4Merck & Co., Inc., Rahway, NJ 07065, USA; 5Department of Surgery, IRCCS Humanitas Research Hospital, 20089 Milan, Italy

**Keywords:** anesthesia, sugammadex, NMBA, rocuronium, residual curarization, neuromuscular blocking agents, surgery

## Abstract

Background: Neuromuscular blocking agent (NMBA) monitoring and reversals are key to avoiding residual curarization and improving patient outcomes. Sugammadex is a NMBA reversal with favorable pharmacological properties. There is a lack of real-world data detailing how the diffusion of sugammadex affects anesthetic monitoring and practice. Methods: We conducted an electronic health record analysis study, including all adult surgical patients undergoing general anesthesia with orotracheal intubation, from January 2016 to December 2019, to describe changes and temporal trends of NMBAs and NMBA reversals administration. Results: From an initial population of 115,046 surgeries, we included 37,882 procedures, with 24,583 (64.9%) treated with spontaneous recovery from neuromuscular block and 13,299 (35.1%) with NMBA reversals. NMBA reversals use doubled over 4 years from 25.5% to 42.5%, mainly driven by sugammadex use, which increased from 17.8% to 38.3%. Rocuronium increased from 58.6% (2016) to 94.5% (2019). Factors associated with NMBA reversal use in the multivariable analysis were severe obesity (OR 3.33 for class II and OR 11.4 for class III obesity, *p*-value < 0.001), and high ASA score (OR 1.47 for ASA III). Among comorbidities, OSAS, asthma, and other respiratory diseases showed the strongest association with NMBA reversal administration. Conclusions: Unrestricted availability of sugammadex led to a considerable increase in pharmacological NMBA reversal, with rocuronium use also rising. More research is needed to determine how unrestricted and safer NMBA reversal affects anesthesia intraoperative monitoring and practice.

## 1. Introduction

Neuro-muscular blocking agents (NMBAs) are among the most commonly employed agent during general anesthesia to facilitate surgical and intubating conditions [1,2]. Despite the development of new molecules and better antagonists, residual postoperative curarization (PORC) remains a significant problem that increases perioperative morbidity and mortality [3,4].

National and international guidelines do not recommend a specific NMBA, suggesting basing the choice of the correct molecule according to patient-specific and surgical characteristics, and advising employing neuro-muscular-transmission (NMT) monitoring [5,6]. Nonetheless, real-world practice is still heterogeneous in monitoring residual curarization [7]. In a sample of 1440 Italian anesthesiologists from 2012, 50% declared using only clinical evaluation for residual curarization, 41% clinical evaluation plus instrumental monitoring, and 9% instrumental monitoring only [7]. Consequently, residual neuromuscular block remains common, and PORC can be detected in up to 60% of patients after surgery, even after NMBA reversal with neostigmine [2].

The advent of sugammadex, a selective NMBA binding agent, for the reversal of rocuronium and vecuronium, provided anesthesiologists with a new tool with the potential of a profound impact on the incidence of PORC [4,5]. In clinical practice, the initial diffusion of sugammadex was restricted due to higher costs. However, in more recent years, a progressive increase in its use has been reported after its availability became unrestricted. This is because of the higher costs associated with sugammadex [4,8].

However, there is a lack of real-world and national registers data detailing how the diffusion of sugammadex in recent years has modified anesthesiologic practice on the use of NMBA and NMBA reversal agents.

We hypothesized that the introduction of sugammadex had a significant impact on intraoperative and perioperative anesthesiologic practice, with significant changes in the choice of NMBA and NMBA reversal.

## 2. Materials and Methods

We conducted a single-center, retrospective observational study, including all adult patients admitted to Humanitas Research Hospital, a large university institute placed in Milan (Italy), between 1 January 2016 and 31 December 2019, for non-cardiac surgery under general anesthesia. The study received IRB approval (no. 757/20) from the Humanitas Research Hospital Independent Ethical Committee. Patient written informed consent was waived for this study by the IRB due to its retrospective design and anonymization of data.

The primary objective of the study was to describe real-world practice patterns of NMBAs and NMBA reversals in adult non-cardiac surgery patients. The secondary objective was to describe the factors associated with NMBA reversal and reversal agents.

We included all adult patients undergoing non-cardiac surgery with general anesthesia with endotracheal intubation, and treated with a NMBA (rocuronium, cisatracurium, atracurium, succinylcholine). Exclusion criteria were history of myasthenia gravis or home therapy with pyridostigmine, end-stage renal failure, use of two types of NMBA reversals (both sugammadex and neostigmine).

The intraoperative and perioperative practice followed standard anesthesiologic practice. The use of NMBA, neuromuscular monitoring, or NMBA reversal was not restricted during the study and was at the choice of the caring anesthesiologist.

Intraoperative and perioperative data were collected from the Electronic Health Record (EHR) system of our center, a high surgical-volume teaching hospital (about 800 beds hospital, 25,000 surgeries per year). Data were extracted from EHR repository and anonymized before inclusion in the study. Following DACQORD principles, intraoperative data were processed to verify for consistency and outliers, as well as for completeness and redundancy [9].

The study followed the STROBE guidelines [10]. Potential source of bias in the cohort study were identified in selection and reporting bias. Selection biases were considered to have minimal impact in this cohort study. This was due to the nature of EHR analysis data, the study selection criteria, and the large population extracted. Reporting bias may have influenced the quality and completeness of entered data. Some data may be under-reported; however, this type of error is unlikely to shift over time or have significant time trends and is limited by EHR completeness medi-co-legal criteria. We used imputation and sensitivity analyses to detect missing data bias.

Sample size estimation: Assuming a 0.30 mean base proportion across 4 years of rocuronium treatment, 8064 patients per year were needed to obtain a margin of error of 1% with 95% confidence interval.

### Statistical Analysis

Variables were described by frequencies (percentage) or mean (SD) and median (IQR), as appropriate. Univariate associations were tested by Chi-Square test, Chi-Square test for trend in proportions, and Mann-Whitney U test, as appropriate, considering a threshold of 0.05 statistical significance. We built a multivariable logistic regression model to describe factors associated with NBMA reversal, and secondary analysis to detect factors associated with the choice of sugammadex in the subpopulation of patients paralyzed with rocuronium. Before inclusion, missing data were analyzed and plotted for each variable to improve model performance. We used random-forest-based imputation of missing data before including variables in the logistic regression model, weighted by the level of missingness per feature (Appendix A). After imputation, the logistic regression model was built including clinically meaningful variables such as year, surgical specialty, and day-hospital surgery by default, and performing the selection of the other variables through a forward-backward stepwise regression approach using Aikake information criteria (AIC) for variable selection. Repeated k-fold cross validation was employed for internal model validation using 10 folds and 10 random repeats. Model discriminative performance was assessed through ROC analysis after cross-validation. Pooled calibration and precision-recall gain curve, which standardized precision against baseline chances expectations, were calculated. Sensitivity analysis on non-missing data only is reported in Appendix A.

All statistical analyses were performed using R software, version 4.1.1.

## 3. Results

A total of 115,046 surgeries from 110,186 patients were extracted from the Electronic Health Record system (EHR). We excluded 72,692 non-cardiac surgeries performed without NMBA or with laryngeal masks and 2110 cardiac surgery cases. According to the other exclusion criteria, the final population consisted of 37,882 surgeries involving 34,571 patients over 4 years. The study flow chart is reported in Figure 1.

Males comprised 44.5% (16,256) of the population, and the median age was 57 (IQR 45–69) years. There were no significant variations in population baseline characteristics over 4 years, while the number of surgeries progressively increased from 8350 in 2016 to 10,376 in 2019. Abdominal surgery accounted for almost one-third of the procedures, followed by neurosurgery (22%), thoracic surgery (15%), and uro-gynecological procedures (14.5%). Hypertension was the most common baseline comorbidity (47.2% of cases), followed by diabetes (9.3%) and chronic obstructive pulmonary disease (COPD) (8.0%).

Most procedures were elective, with urgent/emergency procedures accounting for 5.2% of the total in 2016 and decreasing to 3.6% in 2019. Detailed numbers of interventions in different types of surgery are reported in Appendix A. Inhaled anesthesia was largely prevalent compared to total intravenous anesthesia (TIVA), used in 15% of surgeries, more frequently in day-surgery patients (27% vs. 14%, *p* < 0.0001).

A total of 24,583 (64.9%) were treated with NMBA reversal and 13,299 (35.1%) were not treated with NMBA reversal in the overall population. No adverse events following administration of NMBA reversals (e.g., neostigmine or sugammadex) were reported in the study population. When comparing patients with and without NMBA reversal, the median age was significantly lower in patients treated with reversal [52 (IQR 41–66)], compared to spontaneous reversal [57 (IQR 45–69)].

Higher American Society of Anesthesiology (ASA) scores were more frequent in patients receiving NMBA reversal compared to spontaneous reversal [ASA 3–4 score: 4918 (37.6%) vs. 4311 (18.5%), *p* < 0.001]. Obese patients were more frequent in patients receiving NMBA reversal than in patients receiving spontaneous reversal [class II–III obesity 3923 (30.1%) vs. class II 767 (3.3%, *p* < 0.001)]. The proportion of day-hospital patients was higher in NMBA pharmacological reversal than in spontaneous reversal [850 (6.6%) vs. 720 (3.1%, *p* < 0.001)]. Total intravenous anesthesia was more frequent in spontaneous reversal patients than in pharmacological reversal [4056 (16.5%) vs. 1698 (12.8%), *p* < 0.001)]. Data are reported in Table 1 and graphically represented in Figure 2.

Figure 3 reports a clear time-trend over 4 years in the use of NMBAs reversal (from 25.5% in 2016 to 42.5% in 2019), in the use of sugammadex (from 17.8% in 2016 to 38.3% in 2019), and in the type of NMBA. Rocuronium increased from 58.4% to 94.4% of surgeries.

Time-trend results are reported in Table 2. Other baseline characteristics, including age and gender, did not have a clinically meaningful difference over time. Baseline characteristics and data by year for patients on rocuronium are reported in Appendix A.

The shift toward rocuronium use was more significant in obese patients. While being already used at 95% in 2016 in obesity class III patients, it rose from 51% to 94.0% in obesity class I patients in 4 years. Similarly, patients with respiratory problems such as COPD increased from 43.9% to 77.4% in 2017 to 93.0% in 2019. Rocuronium use in patients undergoing urgent surgeries rose from 29.8% in 2016 to 93% in 2019.

Table 3 reports the results of the multivariable analysis on the factors affecting the need for decurarization at the end of surgery. Multivariable analysis confirms a significant trend over time in the use of decurarization, when controlling for other variables, with odds ratio for pharmacological decurarization increasing by 24% (14–34%) in 2017, 47% (36–59%) in 2018, and 59% (47–72%) in 2019, as compared with 2016 (*p* < 0.001). The association with NMBA reversals administration grew stronger with an increase in ASA classes, with OR 1.37 (1.24–1.52, *p* < 0.001) in ASA 3. NMBA reversals were used more frequently in obese patients, with an increasing trend with increasing body mass index (BMI) status (from OR 1.16 [1.09–1.23], *p* < 0.001] in overweight patients to OR 11.1 [9.81–13.90], *p* < 0.001, in class III obese patients). 

Asthma, Obstructive Sleep Apnea Syndrome (OSAS), and other respiratory comorbidities increased the chances of pharmacological reversal by 18% (3–35%), 19% (10–30%), and 11% (0.1–23%), respectively. General/abdominal surgery was the type of surgery most frequently associated with need for reversal (OR 2.75 [2.35–3.22], *p* < 0.001), even after controlling for rocuronium and cisatracurium dose per kg of body weight OR 3.22 (2.95–3.51) and OR 2.7 (2.0–3.85), respectively. ROC analysis for the model was 0.81 and is reported in Appendix A along with the calibration and other data on model performance.

Secondary analysis on the choice on administering sugammadex in patients receiving rocuronium is reported in Appendix A and identified similar time-trends from 2016 to 2019 and the importance of obesity classes. Model performance is reported in Appendix A, while sensitivity analysis on non-missing data only yielded comparable results to the overall population and is reported in Appendix A.

## 4. Discussion

In this study, we examined a considerable cohort of 37,882 general anesthesia cases over a period of 4 years to assess variations in spontaneous and pharmacological NMBA use and NMBA reversal over time, and to identify real-world factors associated with these patterns.

We demonstrated a significant shift in pharmacological vs. spontaneous reversal, with pharmacological reversal doubling from 25.5% to 42.5% over 4 years. Sugammadex was responsible for most of the increase in pharmacological NMBA antagonism (from 17% in 2016 to 38% in 2019), which was accompanied by a steady decline spontaneous recovery (from 74% to 57%). Additionally, rocuronium usage progressively increased and eventually emerged as the NMBA’s drug of choice.

This change may be interpreted in different ways. National and international guidelines recommend neuromuscular monitoring during surgery, specifically at the end of the surgery before extubation [5,6]. Increased awareness of current guidelines [6,11] on NMT monitoring [12] and total decurarization may explain the rise in NMBA reversal use [13]. On the other hand, this may also be attributed to the emergence of sugammadex, which has a superior safety profile compared to previous anticholinergic drugs [14,15]. Sugammadex was introduced in our center between 2014 and 2015, and gradually became unrestricted in clinical practice thereafter.

Martini et al. recently described similar patterns in a cohort of patients in the Netherlands. In their analysis, unrestricted availability of sugammadex made it progressively the reversal agent preferred by anesthesiologists (up to 99% of cases), and made rocuronium the preferred choice among muscle relaxants (88% of cases) [16]. In addition, a recent study reporting inpatient procedures in the United States found parallel outcomes, with a progressive drop in neostigmine use and a decline in spontaneous recovery over time, following the introduction of unrestricted use of sugammadex [17].

Other intraoperative variables may be considered responsible for this shift. Rapid reversibility of deep block may be advantageous in the first phase of anesthesia when unexpectedly difficult airways are encountered. It may also be advantageous during surgery (by increasing surgical exposure and decreasing insufflation pressure in laparoscopy) and at the conclusion of the procedure, when a deeper block may be required to facilitate surgical closure. Given the option of a deep block reversal, anesthesiologists might have elected to maintain deep NMBA block practically to the conclusion of the surgery, as evidenced in our data by an increase in total rocuronium dose per procedure over time (Table 2). Better operating conditions might minimize theatre occupation time and avoid the increase of general anesthesia or opioid medicines necessary towards the conclusion of surgery when musculoskeletal block wears off, an approach that may be harmful especially for senior patients. [18] The constancy in sugammadex effectiveness and its fewer adverse effects [15] may also counteract the inter-individual variability in rocuronium duration of action observed in prior studies [19], hence enhancing the usage of rocuronium.

As a result, sugammadex may have affected anesthesiologic behavior not only at the end of surgery, but also during the whole intraoperative period. The shift in NMBA choice that we observed in our data confirms the above findings. We observed a significant increase in rocuronium use, from 58.6% in 2016 to 94.5% in 2019, and a corresponding decrease in the use of cisatracurium, atracurium, and suxamethonium. Even though cisatracurium was still used in older patients in our population in 2019 because it has a good pharmacological profile in frail populations with reduced hepatic and liver metabolism [20], anesthesiologists preferred rocuronium, even in frailer patients, because of its prompt reversal [21].

Obesity was among the factors associated with the use of NMBA reversal, with obesity class III patients showing 11 times higher chance of being reversed than underweight or normally weighing patients, a finding confirmed by previous literature [22]. We observed a 37% increase in the use of NMBA among ASA III and ASA IV patients, even when controlling for other factors. The latter may be justified to lower the risk of PORC and possible respiratory complications in a frailer population. Although ASA classification has several limits, ASA class III and IV are the most widely used proxies of increased perioperative risk [23]. The fact that NMBA reversals were used more often in this group supports the idea that a reliable reversal can prevent PORC, even when organ metabolism is lower, as it is in frail patients [24]. Respiratory comorbidities were one of the primary factors driving the anesthesiologist’s choice toward NMBA reversal. Asthma, OSAS, and other respiratory comorbidities were significantly associated with NMBA reversal at the end of surgery [25]. As the majority of the symptoms of PORC are respiratory, the reversal can be explained by the necessity to protect this group from the harmful respiratory consequences of an incomplete NMBA reversal.

Some of the consequence of the increased use of sugammadex in the everyday anesthesiologist’s activity require further consideration. Several articles in the medical literature highlight the risk of hypersensitivity, bradycardia, and hypotension [26,27] following administration of sugammadex, including anecdotical case reports that describe the potential evolution of this clinical picture into asystole [28,29]. The exact mechanism by which sugammadex may induce these hemodynamic impairments remains unknown. These adverse effects have been reported in literature, and the manufacturers of sugammadex identify them as a potential side effect [30,31]. However, since its introduction, adverse effects have remained uncommon and comparable to those associated with other NMBA reversals. In a recent article published by Ju et al. [32], the estimated incidence of sugammadex-induced anaphylaxis was 0.0014%. Furthermore, compared to neostigmine, sugammadex appears to have a decreased incidence of bradycardia in both children and adults [15,33]. Nonetheless, anesthesiologists should constantly evaluate the possibility of hemodynamic abnormalities during and after sugammadex delivery, especially in patients taking heart rate-altering medications.

This observational research produced from EHR data has the limitations inherent to observational studies. Not all data that would be useful were registered in the EHR; nevertheless, we believe this occurrence was uncommon in our cohort. No adverse events were recorded following the administration of sugammadex or neostigmine to any of the trial participants. However, it is possible that this information was not always included in the EHR data, leading to underreporting of adverse effects in this cohort.

No data on the occurrence of major complications related with PORC were reported in our cohort. While major events are uncommon, this may also be attributable to underreporting, and we cannot rule out the possibility that some patients transferred to the ICU after surgery experienced issues associated with residual curarization. In addition, we did not gather data on aspiration pneumonia or later complications that become evident later over the course of the whole hospital stay.

Sensitivity analyses were conducted to detect and describe the risk of reporting bias, and we checked for discrepancies and misclassification errors. EHR did not register consistent data on monitoring of depth of NMBA block during surgery, which would be helpful to better assess the indication for reversal, even if NMT monitoring is part of standard practice and enforced by internal clinical protocol. Thus, our data reflect the need for a reversal at the end of surgery and the choice of the caring anesthesiologist. Since data for this study include a single site in Italy, results may have limited generalizability and may not reflect global practice.

## 5. Conclusions

We demonstrated a significant increase of NMBA reversal use over 4 years, which appears to be driven mainly by the recent increase in availability of sugammadex. Obesity, ASA higher class, and respiratory comorbidities were the factors most associated with sugammadex reversal. There was a parallel increase in the use of rocuronium as the preferred blocking agent over 4 years. Further studies are needed to assess changes in anesthesiology monitoring and practice with the availability of newer and safer drugs.

## Figures and Tables

**Figure 1 jcm-12-00563-f001:**
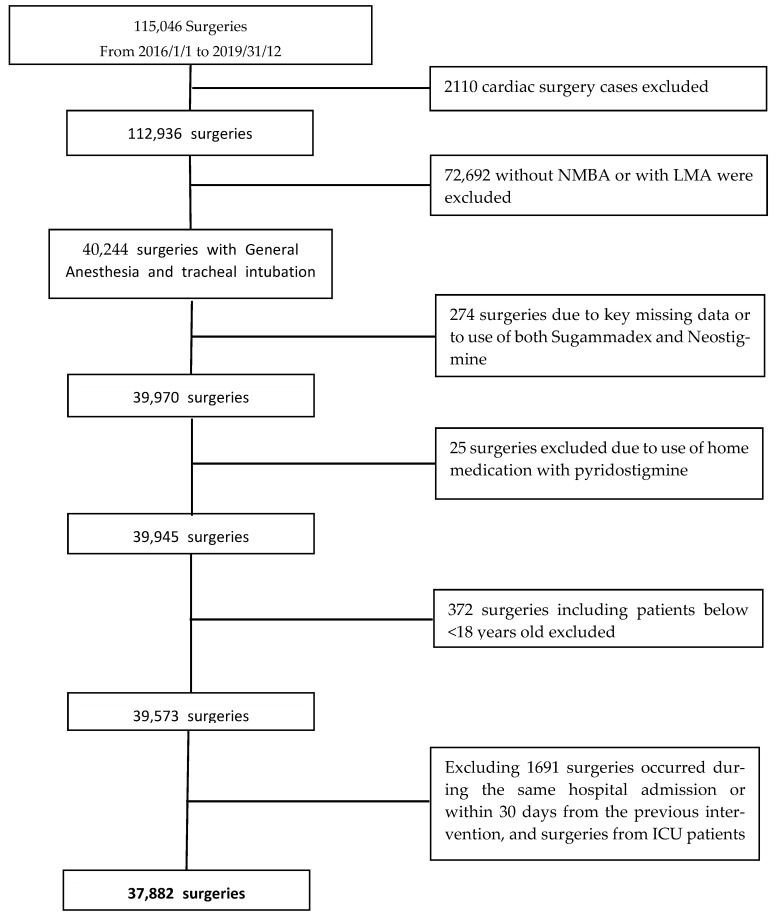
Flow chart of patient selection process.

**Figure 2 jcm-12-00563-f002:**
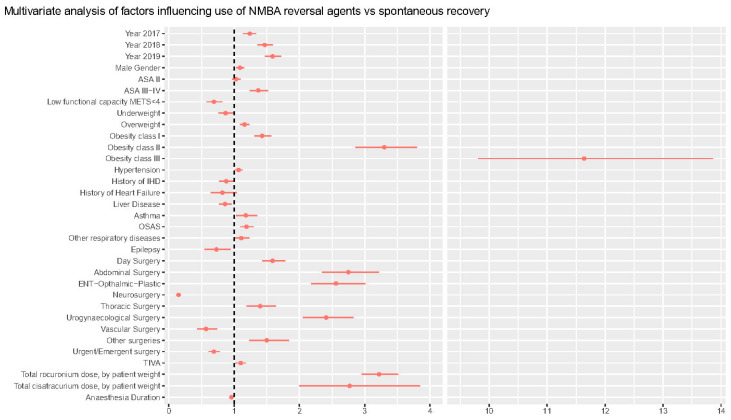
Time trends in NMBA reversal practice.

**Figure 3 jcm-12-00563-f003:**
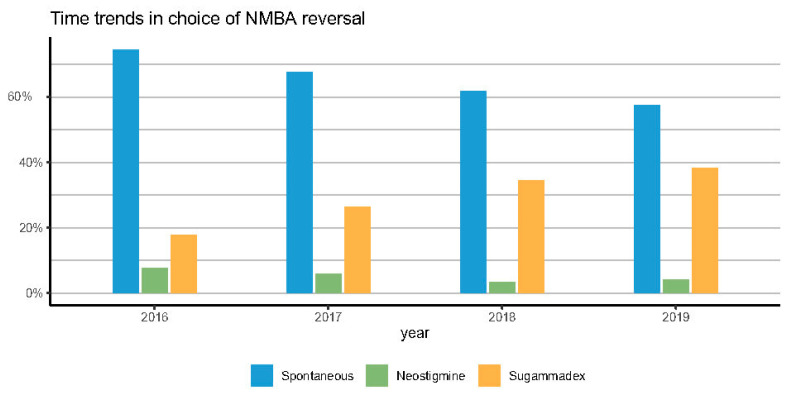
Forest plot of multivariable analysis.

**Table 1 jcm-12-00563-t001:** Demographics and baseline characteristics of the NMB reversal population vs. the NMB non reversal population.

	Overall (37,882)	Spontaneous Reversal (24,583; 64.9%)	Pharmacological Reversal (13,299; 35.1%)	*p*-Value
Age (continuous)	57 (45–69)	59 (47–70)	52 (41–66)	<0.001
Age bins				
18–40	6354 (17.4%)	3347 (14.2%)	3007 (23.3%)	<0.001
40–50	6987 (19.1%)	4149 (17.6%)	2838 (22.0%)	
50–60	7766 (21.3%)	5034 (21.3%)	2732 (21.1%)	
60–70	7791 (21.3%)	5533 (23.5%)	2258 (17.5%)	
70–80	6214 (17.0%)	4549 (19.3%)	1665 (12.9%)	
80–90	1391 (3.8%)	973 (4.1%)	418 (3.2%)	
Male	16,256 (44.5%)	11,042 (46.8%)	5214 (40.4%)	<0.001
ASA class				
I	9667 (26.5%)	6865 (29.4%)	2802 (21.4%)	<0.001
II	17,550 (48.2%)	12,183 (52.2%)	5367 (41.0%)	
III	9099 (25.0%)	4219 (18.1%)	4880 (37.3%)	
IV	130 (0.4%)	92 (0.4%)	38 (0.3%)	
BMI class				
Underweight (<18.5)	1384 (3.8%)	945 (4.1%)	439 (3.4%)	<0.001
Normal weight (18.5–24.9)	15,555 (42.8%)	11,034 (47.4%)	4521 (34.7%)	
Overweight (25–29.9)	11,164 (30.7%)	8076 (34.7%)	3088 (23.7%)	
Obesity class I (30–34.9)	3515 (9.7%)	2440 (10.5%)	1075 (8.2%)	
Obesity class II (35–39.9)	1556 (4.3%)	538 (2.3%)	1018 (7.8%)	
Obesity class III (>40)	3134 (8.6%)	229.0 (1.0%)	2905 (22.3%)	
Allergy	12,081 (33.1%)	7555 (32.3%)	4526 (34.6%)	<0.001
Smoker	23,925 (74.3%)	14,960 (74.3%)	8965 (74.3%)	0.9
METS < 4	925 (2.7%)	663 (3.1%)	262 (2.1%)	<0.001
Comorbidities				
Arterial hypertension	12,037 (47.2%)	7892 (47.2%)	4145 (47.2%)	>0.9
Arrhythmias	1993 (7.8%)	1414 (8.5%)	579 (6.6%)	<0.001
Vascular disease	1294 (5.1%)	988 (5.9%)	306 (3.5%)	<0.001
Ischemic heart disease	1652 (6.5%)	1184 (7.1%)	468 (5.3%)	<0.001
Heart Failure	442 (1.7%)	290 (1.7%)	152 (1.7%)	>0.9
Other cardiologic comorbidities	1728 (6.8%)	1119 (6.7%)	609 (6.9%)	0.5
Liver disease	882 (2.7%)	680 (3.2%)	202 (1.7%)	<0.001
Chronic Kidney failure	779 (2.3%)	543 (2.5%)	236 (2.0%)	0.003
Asthma	1316 (4.2%)	711 (3.3%)	605 (6.2%)	<0.001
COPD	2520 (8.1%)	1747 (8.2%)	773 (7.9%)	0.4
OSAS	925 (3.0%)	267 (1.2%)	658 (6.7%)	<0.001
Other respiratory comorbidities	2816 (9.0%)	1426 (6.7%)	1390 (14.2%)	<0.001
Epilepsy	570 (1.7%)	472 (2.2%)	98 (0.9%)	<0.001
Stroke	762 (2.3%)	524 (2.4%)	238 (2.1%)	0.038
Dementia	205 (0.6%)	150 (0.7%)	55 (0.5%)	0.017
Surgical characteristics				
Day surgery	1570 (4.3%)	720 (3.1%)	850 (6.6%)	<0.001
Type of surgery				
General/Abdominal surgery	10,932 (28.9%)	4719 (19.2%)	6213 (46.7%)	<0.001
ENT-Ophthalmic-Plastic	4772 (12.6%)	2511 (10.2%)	2261 (17.0%)	
Neurosurgery	7960 (21.0%)	7608 (30.9%)	352 (2.6%)	
Orthopedic surgery	1144 (3.0%)	897 (3.6%)	247 (1.9%)	
Other surgeries	1099 (2.9%)	730 (3.0%)	369 (2.8%)	
Thoracic surgery	6181 (16.3%)	4336 (17.6%)	1845 (13.9%)	
Urologic and gynecologic surgery	5007 (13.2%)	3111 (12.7%)	1896 (14.3%)	
Vascular surgery	787 (2.1%)	671 (2.7%)	116 (0.9%)	
Urgent\Emergency surgery				
Elective surgery	34,367 (95.5%)	21,952 (94.9%)	12,415 (96.6%)	<0.001
Urgent\Emergency surgery	1627 (4.5%)	1189 (5.1%)	438 (3.4%)	
Type of Anaesthesia				
Inhaled	32,128 (84.8%)	20,527 (83.5%)	11,601 (87.2%)	<0.001
TIVA	5754 (15.2%)	4056 (16.5%)	1698 (12.8%)	
Rocuronium (median-IQR dose)	30,896 (81.6%)	21.8 mg (14.2)	35.3 (54.1)	
Cisatracurium (median-IQR dose)	6710 (17.7%)	16 (8)	18 (8)	
Length of surgery (min)	120 (80, 190)	134 (90, 200)	95 (64, 175)	<0.001
Year of enrolment				
2016	8354 (22.1%)	6225 (25.4%)	2129 (16.0%)	<0.001
2017	9026 (23.9%)	6105 (24.9%)	2921 (22.0%)	
2018	10,011 (26.5%)	6197 (25.3%)	3814 (28.7%)	
2019	10,376 (27.5%)	5970 (24.4%)	4406 (33.2%)	

TIVA = total intravenous anesthesia, COPD = chronic Obstructive Pulmonary Disease, OSAS = Obstructive Sleep Apnea Syndrome. MET = Metabolic equivalent of tasks.

**Table 2 jcm-12-00563-t002:** Time trends of NMBA reversal, NMBA use, and patient and surgical characteristics over 4 years.

	Total ^1^	By Year ^1^					
Characteristic	N = 37,882	2016, N = 8354	2017, N = 9026	2018, N = 10,011	2019, N = 10,376	*p*-Value ^2^	*p*-for-Trend
Pharmacological reversal (NMBRA)	13,299.0 (35.1%)	2129.0 (25.5%)	2921.0 (32.4%)	3814.0 (38.1%)	4406.0 (42.5%)	<0.001	<0.001
Type of NMBRA							
Neostigmine	1962.0 (5.2%)	642.0 (7.7%)	528.0 (5.8%)	348.0 (3.5%)	434.0 (4.2%)	<0.001	<0.001
Sugammadex	11,337.0 (29.9%)	1487.0 (17.8%)	2393.0 (26.5%)	3466.0 (34.6%)	3972.0 (38.3%)		
Spontaneous reversal	24,583.0 (64.9%)	6225.0 (74.5%)	6105.0 (67.6%)	6197.0 (61.9%)	5970.0 (57.5%)		
Type of NMBA							
Rocuronium	30,896.0 (81.6%)	4881.0 (58.4%)	7121.0 (78.9%)	9013.0 (90.0%)	9797.0 (94.4%)	<0.001	<0.001
Cisatracurium	6710.0 (17.7%)	3301.0 (39.5%)	1843.0 (20.4%)	974.0 (9.7%)	563.0 (5.4%)		
Other NMBA	276.0 (0.7%)	172.0 (2.1%)	62.0 (0.7%)	24.0 (0.2%)	16.0 (0.2%)		
Age	57 (45, 69)	57 (45, 69)	56 (45, 69)	57 (45, 69)	57 (45, 69)	0.5	
Age bins							
18–40	6354.0 (17.4%)	1339.0 (17.4%)	1467.0 (17.1%)	1717.0 (17.3%)	1831.0 (17.8%)	0.002	0.9
40–50	6987.0 (19.1%)	1506.0 (19.6%)	1720.0 (20.0%)	1808.0 (18.2%)	1953.0 (19.0%)		
50–60	7766.0 (21.3%)	1534.0 (20.0%)	1789.0 (20.8%)	2196.0 (22.1%)	2247.0 (21.8%)		
60–70	7791.0 (21.3%)	1716.0 (22.3%)	1855.0 (21.6%)	2079.0 (20.9%)	2141.0 (20.8%)		
70–80	6214.0 (17.0%)	1315.0 (17.1%)	1459.0 (17.0%)	1735.0 (17.5%)	1705.0 (16.6%)		
80–90	1391.0 (3.8%)	279.0 (3.6%)	301.0 (3.5%)	394.0 (4.0%)	417.0 (4.1%)		
Male	16,256.0 (44.5%)	3512.0 (45.7%)	3745.0 (43.6%)	4389.0 (44.2%)	4610.0 (44.8%)	0.049	0.51
ASA							
II	17,550.0 (48.2%)	3923.0 (51.4%)	4096.0 (46.8%)	4604.0 (47.1%)	4877.0 (47.9%)	<0.001	<0.001
I	9667.0 (26.5%)	2030.0 (26.6%)	2349.0 (26.8%)	2605.0 (26.7%)	2651.0 (26.0%)		
III	9099.0 (25.0%)	1661.0 (21.8%)	2267.0 (25.9%)	2531.0 (25.9%)	2620.0 (25.7%)		
IV	130.0 (0.4%)	21.0 (0.3%)	43.0 (0.5%)	33.0 (0.3%)	32.0 (0.3%)		
BMI class							
Underweight (<18.5)	1384.0 (3.8%)	279.0 (3.7%)	347.0 (4.0%)	389.0 (4.0%)	366.0 (3.6%)	0.004	0.042
Normal weight (18.5–24.9)	15,555.0 (42.8%)	3247.0 (42.6%)	3712.0 (42.5%)	4233.0 (43.5%)	4321.0 (42.6%)		
Overweight (25–29.9)	11,164.0 (30.7%)	2386.0 (31.3%)	2713.0 (31.1%)	2942.0 (30.2%)	3078.0 (30.4%)		
Obesity class I (30–34.9)	3515.0 (9.7%)	782.0 (10.3%)	842.0 (9.7%)	950.0 (9.8%)	932.0 (9.2%)		
Obesity class II (35–39.9)	1556.0 (4.3%)	308.0 (4.0%)	388.0 (4.4%)	408.0 (4.2%)	448.0 (4.4%)		
Obesity class III (>40)	3134.0 (8.6%)	617.0 (8.1%)	722.0 (8.3%)	807.0 (8.3%)	988.0 (9.8%)		
History of allergy	12,081.0 (33.1%)	2441.0 (32.0%)	2953.0 (33.7%)	3305.0 (33.8%)	3356.0 (33.0%)	0.042	0.25
Smoking	8281.0 (25.7%)	1751.0 (26.1%)	1994.0 (26.2%)	2221.0 (25.6%)	2291.0 (25.2%)	0.4	0.114
Low METS (<4)	925.0 (2.7%)	179.0 (3.3%)	307.0 (3.5%)	221.0 (2.3%)	214.0 (2.1%)	<0.001	<0.001
Hypertension	12,037.0 (47.2%)	2554.0 (70.4%)	2942.0 (43.1%)	3217.0 (42.8%)	3284.0 (44.1%)	<0.001	<0.001
History of arrhythmia	1993.0 (7.8%)	343.0 (9.5%)	480.0 (7.0%)	576.0 (7.7%)	585.0 (7.9%)	<0.001	<0.001
Vasculopathy	1294.0 (5.1%)	232.0 (6.4%)	310.0 (4.5%)	355.0 (4.7%)	392.0 (5.3%)	<0.001	0.2
Ischemic heart disease	1652.0 (6.5%)	381.0 (10.5%)	398.0 (5.8%)	463.0 (6.2%)	397.0 (5.3%)	<0.001	<0.001
Heart failure	442.0 (1.7%)	88.0 (2.4%)	90.0 (1.3%)	137.0 (1.8%)	125.0 (1.7%)	<0.001	0.2
Other cardiac disease	1728.0 (6.8%)	303.0 (8.4%)	360.0 (5.3%)	506.0 (6.7%)	555.0 (7.4%)	<0.001	0.34
Liver disease	882.0 (2.7%)	141.0 (2.0%)	232.0 (2.9%)	228.0 (2.6%)	278.0 (3.0%)	<0.001	0.002
Chronic Kidney disease	779.0 (2.3%)	154.0 (2.2%)	177.0 (2.2%)	235.0 (2.6%)	210.0 (2.3%)	0.2	0.47
Asthma	1316.0 (4.2%)	275.0 (4.1%)	280.0 (3.7%)	356.0 (4.3%)	404.0 (4.7%)	0.015	0.01
COPD	2520.0 (8.1%)	515.0 (7.6%)	601.0 (7.9%)	685.0 (8.4%)	711.0 (8.3%)	0.3	0.1
OSAS	925.0 (3.0%)	174.0 (2.6%)	198.0 (2.6%)	249.0 (3.0%)	304.0 (3.5%)	<0.001	<0.001
Other respiratory disease	2816.0 (9.0%)	604.0 (8.9%)	599.0 (7.9%)	709.0 (8.7%)	894.0 (10.4%)	<0.001	<0.001
Epilepsy	570.0 (1.7%)	133.0 (1.9%)	146.0 (1.8%)	154.0 (1.7%)	135.0 (1.5%)	0.2	0.045
Stroke	762.0 (2.3%)	156.0 (2.2%)	214.0 (2.7%)	185.0 (2.1%)	204.0 (2.3%)	0.055	0.47
Dementia	205.0 (0.6%)	41.0 (0.6%)	52.0 (0.7%)	58.0 (0.7%)	53.0 (0.6%)	0.9	0.9
Day surgery	1570.0 (4.3%)	190.0 (2.5%)	296.0 (3.4%)	498.0 (5.0%)	586.0 (5.7%)	<0.001	<0.001
Type of surgery							
ENT-Ophthalmic-Plastic	4772.0 (12.6%)	907.0 (10.9%)	923.0 (10.2%)	1294.0 (12.9%)	1642.0 (15.8%)	<0.001	0.117
Neurosurgery	7960.0 (21.0%)	1888.0 (22.6%)	1938.0 (21.5%)	2059.0 (20.6%)	2038.0 (19.6%)		
Thoracic surgery	6181.0 (16.3%)	1277.0 (15.3%)	1484.0 (16.4%)	1688.0 (16.9%)	1721.0 (16.6%)		
Abdominal surgery	10,932.0 (28.9%)	2418.0 (28.9%)	2605.0 (28.9%)	2883.0 (28.8%)	3002.0 (28.9%)		
Orthopedic surgery	1144.0 (3.0%)	303.0 (3.6%)	280.0 (3.1%)	270.0 (2.7%)	279.0 (2.7%)		
Uro-gynecological surgery	5007.0 (13.2%)	1208.0 (14.5%)	1265.0 (14.0%)	1296.0 (12.9%)	1221.0 (11.8%)		
Vascular surgery	787.0 (2.1%)	220.0 (2.6%)	227.0 (2.5%)	196.0 (2.0%)	139.0 (1.3%)		
Other surgery	1099.0 (2.9%)	133.0 (1.6%)	304.0 (3.4%)	325.0 (3.2%)	334.0 (3.2%)		
Elective Surgery	34,367.0 (95.5%)	7142.0 (94.8%)	8173.0 (94.8%)	9255.0 (95.7%)	9702.0 (96.4%)	<0.001	<0.001
Urgent\Emergency Surgery	1627.0 (4.5%)	392.0 (5.2%)	448.0 (5.2%)	416.0 (4.3%)	363.0 (3.6%)		
TIVA	5754.0 (15.2%)	1205.0 (14.4%)	1287.0 (14.3%)	1651.0 (16.5%)	1593.0 (15.4%)	<0.001	0.003
Median rocuronium dose (mg)	60.2 (32.1)	56.3 (32.0)	59.0 (31.0)	60.0 (32.5)	63.2 (32.5)	<0.001	
Median cisatracurium dose (mg)		17.9 (9.0)	18.1 (9.3)	19.8 (20.2)	18.8 (13.8)	<0.001	
Length of surgery	120 (80, 190)	121 (80, 190)	125 (80, 195)	120 (77, 190)	115 (75, 188)	<0.001	

^1^ n (%); Median (IQR), ^2^ Pearson’s Chi-squared test; Kruskal-Wallis rank sum test. TIVA = total intravenous anesthesia, COPD = chronic Obstructive Pulmonary Disease, OSAS = Obstructive Sleep Apnea Syndrome. MET = Metabolic equivalent of tasks.

**Table 3 jcm-12-00563-t003:** Multivariable model to identify factors associated with NMBA reversal.

Characteristic	OR ^1^	95% CI ^2^	*p*-Value
Year (vs. 2016)			
2017	1.24	1.14–1.34	<0.001
2018	1.47	1.36–1.59	<0.001
2019	1.59	1.47–1.72	<0.001
Male Gender	1.09	1.03–1.15	0.003
ASA class (vs. class I)			
Class II	1.03	0.97–1.10	0.3
Class III-IV	1.37	1.24–1.52	<0.001
METS < 4	0.69	0.58–0.82	<0.001
BMI class (vs. Normal weight)			
Underweight (<18.5)	0.87	0.76–0.99	0.03
Overweight (25–29.9)	1.16	1.09–1.23	<0.001
Obesity class I (30–34.9)	1.43	1.31–1.57	<0.001
Obesity class II (35–39.9)	3.30	2.86–3.80	<0.001
Obesity class III (>40)	11.1	9.81–13.9	<0.001
Day surgery vs. inpatient surgery	1.59	1.43–1.78	<0.001
Type of surgery			
Orthopedic surgery (ref)	—	—	
Abdominal surgery	2.75	2.35–3.22	<0.001
ENT-Ophthalmic-Plastic surgery	2.56	2.18–3.01	<0.001
Neurosurgery	0.15	0.13–0.18	<0.001
Thoracic surgery	1.40	1.19–1.64	<0.001
Uro-gynecology surgery	2.41	2.05–2.83	<0.001
Vascular surgery	0.57	0.44–0.74	0.013
Other surgeries	1.50	1.23–1.84	<0.001
Urgent\Emergency surgery	0.69	0.61–0.78	<0.001
TIVA	1.10	1.02–1.18	0.015
Anaesthesia length (per 10-min increase)	0.96	0.96–0.97	<0.001
Total rocuronium dose (on patient weight)	3.22	2.95–3.51	<0.001
Total cisatracurium dose (on patient weight)	2.77	2.00–3.85	<0.001
Hypertension	1.07	1.01–1.13	0.02
History of IHD	0.88	0.77–1.01	0.07
History of heart failure	0.82	0.64–1.04	0.11
Liver disease	0.86	0.77–0.96	0.005
Asthma	1.18	1.03–1.35	0.015
OSAS	1.19	1.10–1.30	<0.001
Other respiratory comorbidities	1.11	1.00–1.23	0.056
Epilepsy	0.73	0.55–0.95	0.022

^1^ Odds Ratio, ^2^ 95% Confidence Interval, TIVA = total intravenous anaesthesia, OSAS = Obstructive Sleep Apnea Syndrome, IHD = Ischemic Heart Disease, MET = Metabolic equivalent of tasks.

## Data Availability

Data are available from the authors upon reasonable request and can be shared only after approval of the hospital research office.

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
