# Peer review of "REVersal of nEuromusculAr bLocking Agents in Patients Undergoing General Anaesthesia (REVEAL Study)"

_jcm, 2023, doi:10.3390/jcm12020563_

Round 1

Reviewer 1 Report

The submitted work complies with the task it pursues and is suitable for publication after some revisions.

In this single-center and retrospective observational study, the authors described real-world practice patterns of NMBAs and NMBA reversal administration in adult non-cardiac surgery patients, analyzing factors that could be associated with preferential use of NMBAs and NMBA reversal agents (sugammadex).

The lack of national registers in Italy makes it difficult to collect and analyze data about the use and diffusion of these drugs. Considering their increasing use and the consequent higher number of patients to which they are administered, the topic is of interest.

I don’t have to propose substantial corrections, however, some revisions could improve the manuscript:

-       The English language needs to be improved. There are several too-long and misleading sentences.

-       It is necessary to insert the name and the localization of the “large university hospital” in the text.

-       Considering the importance of the topic, it should be necessary, if it is possible, to insert and describe data about adverse events after the sugammadex administration. Otherwise, it should be included in the limitations. In Italy, underreporting is a problem of great importance.

-       The increasing adverse events described in the literature and linked to sugammadex should be opportunely discussed. I recommend you read the Editorial that accompanied the BJA issue for one of your references: Hunter and Naguib. Sugammadex-induced bradycardia and asystole: how great is the risk? BJA 2018;121:8-12. This editorial highlights that reports of adverse events for Sugammadex (in relation to serious cardiac events) have far outpaced such reporting for neostigmine, pointing out the necessity of continuous monitoring of sugammadex administration and more rigorous reporting of adverse events to agencies. Your discussion section would be strengthened if you comment on the most recent case reports (e.g. Becerra-Bolanos et al, Teng et al, Fierro et al, etc). briefly describing the complex etiology of the adverse events.

-       It’s better to include the limitations of the study in the discussion without the sub-paragraph.

Reviewer 2 Report

Major comment

 PORC is an important challenge after the NMBA's reversals. Also, you emphasized PORC in the introduction. If you provide additional data that the incidence of PORCU has decreased in PACU for 4 years, it will support the results of the changes in NMBRA derived from this study. Please suggest the frequency of occurrence of POCU for 4 years.

Minor comment  

Please correct the numbers in end of line 56, 59. 

Round 2

Reviewer 2 Report

Your manuscript was completed more logically after the revision.